# Comorbidity and Cancer Disease Rates among Those at High-Risk for Alzheimer’s Disease: A Population Database Analysis

**DOI:** 10.3390/ijerph192416419

**Published:** 2022-12-07

**Authors:** David Valentine, Craig C. Teerlink, James M. Farnham, Kerry Rowe, Heydon Kaddas, JoAnn Tschanz, John S. K. Kauwe, Lisa A. Cannon-Albright

**Affiliations:** 1Department of Biology, Brigham Young University, Provo, UT 84602, USA; 2Genetic Epidemiology, Department of Internal Medicine, University of Utah School of Medicine, Salt Lake City, UT 84132, USA; 3National Oncology Program, Veterans Administration, Durham, NC 27705, USA; 4Department of Psychology, Utah State University, Logan, UT 84322, USA; 5George E. Wahlen Department of Veterans Affairs Medical Center, Salt Lake City, UT 84148, USA; 6Huntsman Cancer Institute, Salt Lake City, UT 84112, USA

**Keywords:** Alzheimer’s disease, family history, UPDB, comorbidity, cancer, hypertension, diabetes, dementia

## Abstract

(1) Importance: Alzheimer’s disease (AD) is complex and only partially understood. Analyzing the relationship between other more treatable or preventable diseases and AD may help in the prevention and the eventual development of treatments for AD. Risk estimation in a high-risk population, rather than a population already affected with AD, may reduce some bias in risk estimates. (2) Objective: To examine the rates of various comorbidities and cancers in individuals at high-risk for AD, but without a clinical diagnosis, relative to individuals from the same population with normal AD risk. (3) Design, Setting, and Participants: We conducted a study using data from the Utah Population Database (UPDB). The UPDB contains linked data from the Utah Cancer Registry, Utah death certificates, the Intermountain Health patient population, and the University of Utah Health patient population. Subjects were selected based on the availability of ancestral data, linked health information, and self-reported biometrics. (4) Results: In total, 75,877 participants who were estimated to be at high risk for AD based on family history, but who did not have an active AD diagnosis, were analyzed. A lower incidence of diabetes (RR = 0.95, 95% CI [0.92,0.97], *p* < 0.001), hypertension (RR = 0.97, 95% CI [0.95,0.99], *p* < 0.001), and heart disease (RR = 0.95, 95% CI [0.93,0.98], *p* < 0.001) was found. There was no difference in rates of cerebrovascular disease or other forms of dementia. Of the 15 types of cancer analyzed: breast (RR = 1.23, 95% CI [1.16, 1.30], *p* < 0.001); colorectal (RR = 1.30, 95% CI [1.21, 1.39], *p* < 0.001); kidney (RR = 1.49, 95% CI (1.29, 1.72), *p* < 0.001); lung (RR = 1.25, 95% CI [1.13, 1.37], *p* < 0.001); non-Hodgkin’s Lymphoma (RR = 1.29, 95% CI [1.15, 1.44], *p* < 0.001); pancreas (RR = 1.34, 95% CI [1.16, 1.55], *p* < 0.001); stomach (RR = 1.59, 95% CI [1.36, 1.86], *p* < 0.001); and bladder (RR = 1.40, 95% CI [1.25, 1.56], *p* < 0.001), cancers were observed in significant excess among individuals at high-risk for AD after correction for multiple testing. (5) Conclusions and Relevance: Since age is the greatest risk factor for the development of AD, individuals who reach more advanced ages are at increased risk of developing AD. Consistent with this, people with fewer comorbidities earlier in life are more likely to reach an age where AD becomes a larger risk. Our findings show that individuals at high risk for AD have a decreased incidence of various other diseases. This is further supported by our finding that our high-risk group was also found to have an increased incidence of various cancers, which also increase in risk with age. There is the possibility that a more meaningful or etiological relationship exists among these various comorbidities. Further research into the etiological relationship between AD and these comorbidities may elucidate these possible interactions.

## 1. Introduction

Alzheimer’s is a disease that increases in frequency with age and occurs in individuals with various co-morbidities. The relationship between AD and other diseases is of particular interest because the co-morbidities may serve as early risk signs for subsequent AD development, may share a treatable pathway or etiology, and may help to slow disease progression in the AD patients [1,2]. Additionally, there may be a protective effect between AD and various cancers, as well as shared pathways that may reveal novel treatment options [3,4,5,6]. Previous studies on this topic have been inconsistent, with some concluding that people with an active Alzheimer’s Disease diagnosis are more likely to have co-morbidities, such as hypertension [7,8,9], heart disease [10,11,12], diabetes [7,10,13], cerebrovascular disease [7,11], and other dementias [11,14,15], but several others finding no relationship [16,17,18]. Similarly, studies that examine possible interplay between AD and various cancers have had mixed results with some finding people with AD to be less likely to have certain cancers [19,20,21], but others showing no relationship. Further, these studies have failed to account for the increased morbidity in both diseases, or have found that the cancer treatment may increase the risk for subsequent AD development [22,23]. Prior inconsistent results were largely from experimental surveys or animal models. Others were based on active Alzheimer’s diagnosis or autopsy, making it difficult to control for the myriad of complications that come from an active AD diagnosis and the wide variety of treatment and lifestyle and the effects on disease progression. Additionally, analyzing relationships between AD and comorbid conditions based on active Alzheimer’s diagnosis introduces the difficulty in calculating time-at-risk and the statistical complications in controlling for it. Fortunately, advances in database access, linking, and analysis enable new population and sample definitions and analyses. These new analyses remove the time-at-risk problem, as well as the complications of the spectrum of AD progression, allowing new insight into the relationships between AD and various co-morbidities.

In this study, we analyzed participants in from the Utah Population Database (UPDB), a repository of information for approximately 11 million individuals, with more than 1.8 million of these individuals having extensive genealogical information and linked medical records. Using risk estimates for specific family history constellations for AD adapted from a previous study [24], we identified 85,787 individuals at high risk for AD, but who have not been diagnosed with AD. These selection criteria avoid the potential confounding factors that an active AD diagnosis bring, as well as help remove the complications of time-at-risk calculations. These individuals were analyzed in sex, 5-year birth year, and birth state matched cohorts and expected numbers of affected subjects were estimated using cohort-specific rates of comorbid conditions (hypertension, diabetes, heart disease, cerebrovascular disease, other dementias, and cancer) estimated in the UPDB population with linked genealogy and medical data.

## 2. Methods

### 2.1. Sample and Study Design

The UPDB is a computerized genealogy of the Utah founders and their descendants, starting in the 1800s and continuing through to the present. The UPDB currently contains various records for approximately 11 million individuals, including extensive genealogy for over 1.8 million subjects. These genealogies include data for at least 12 of their 14 immediate ancestors (both parents, all 4 grandparents, and at least 6 of 8 great grandparents). These individuals with extensive and uniform genealogic data were analyzed here.

The UPDB is record-linked to several key datasets, including the Utah Cancer Registry (1966–2014), Utah death certificates (1904–2014), and electronic medical records (EMR) from the two largest healthcare providers in Utah: Intermountain Healthcare (IM) and the University of Utah Health Sciences Center (UUHSC) (1994–2014). Of the 1.8 million individuals with extensive genealogy, there are 207,612 individuals with linked death certificate data; 135,814 individuals with linked cancer registry data; 934,437 individuals with linked Drivers’ License data; and 1,186,823 individuals with a linked EMR from IM or UUHSC. Cause of death data on Utah death certificates was coded with ICD revisions 6–10 and includes either primary cause of death or all contributing causes of death, depending on the year of death. Diagnosis coding in IM- and UUHSC-linked EMR is in ICD revisions 9 and 10. Utah Drivers’ License data have been linked from 1972 and includes the date of new or renewal issue and self-reported height and weight.

The UPDB represents the historical and current population of Utah. Utah was largely founded by Northern Europeans [25] and the population today is still reported as 85% white. The population of the state was 11,380 in the 1850 census, 40,273 in the 1860 census and is now over 3 million, with the youngest population in the US at a median age of 30.3 years; females constitute 49.7% of the state. The state has the highest literacy rate in the US [26].

### 2.2. Identification of Individuals at High-Risk for AD

We identified the set of subjects estimated to be at high risk of AD (Relative Risk > 2.0) by their complete and extended family history for AD, as described previously and summarized below. All AD cases with extensive linked genealogy data as described were identified using linked Utah Death Certificates and healthcare data. Over 100 various family history constellations for AD were considered, based on the number of first-degree, second-degree, and third-degree relatives affected with AD, and maternal or paternal inheritance [24]. All individuals with each specific family history constellation were identified in UPDB and considered as probands for estimation of AD risk. The risk of Alzheimer’s disease for the set of all probands with each family history constellation was estimated as the observed number of probands diagnosed with AD divided by the expected number of probands diagnosed with AD. The observed number of AD-affected probands for each constellation was counted; the expected number of AD-affected probands was estimated by summing the cohort-specific rate of AD for each proband. The cohort-specific rate of AD was calculated by assigning all 1.8 million individuals with extensive genealogy data to a sex, 5-year-birth year, and birth state (Utah or not) cohort. The cohort specific rate for AD was estimated as the number of AD cases in each cohort, divided by the number of UPDB individuals in each cohort. All individuals belonging to any family history constellation with estimated Relative Risk for AD > 2.0 were considered at high risk for Alzheimer’s disease. In total, 87,605 individuals at high-risk for AD based on family history of AD were identified; the 1818 who were already diagnosed with AD were removed from this set of individuals to eliminate any confounding due to the AD diagnosis. In total, 85,787 individuals were at high-risk for AD based on family history of AD, but not diagnosed with AD were analyzed here.

### 2.3. Definition of Phenotypes to Be Tested for Association

To identify individuals diagnosed with comorbid conditions, we searched linked EMR data from IM and UUHSC for ICD 9 and ICD 10 codes for AD and all other comorbid conditions considered (diabetes—ICD9: 250, ICD10: E10, E11, E14; hypertension—ICD9: 401–404, ICD10: I10–I13, I270; heart disease—ICD9 391, 393–398, 410–414, 424–425, 427.3, 427.4, 427.5, 428, 440, 443.9, ICD10: I01, I02, I05–I09, I21, I22, I24, I34–I37, I45.9, I46–I50, I70, I110, I130, I132, I25, I250, I251, I255, I420–I422, I425, I428, I429, I490, I739, Z95.1; cerebrovascular disease: ICD9: 346, ICD10: I63, I67.2, I67.89, I67.9, I69, G43, G46, Z86.73; dementia excluding AD—ICD9: 290, 292.82, 294, 331.11, 331.82, 331.83, 332, ICD10: F01–F03, F18.97, F19.17, F19.97, G20, G31.0, G31.83, G31.84), and cancer, for which ICD-Oncology NCI SEER primary site coding was used. BMI was calculated using the self-reported height and weight recorded in the most recent linked Utah Drivers’ License data.

### 2.4. Comorbid Phenotype Disease Rates

Population disease rates for each comorbidity considered were required for calculation of Relative Risk (RR). All individuals with a linked EMR from UUHSC (*n* = 323,277) or IM (*n* = 1,706,220) were assigned to a sex, 5-year birth year, and birth state cohort (in or out of Utah). Cohort-specific rates were estimated as the number of individuals diagnosed with the specific comorbid phenotype (e.g., diabetes) divided by the total number of individuals assigned to the cohort.

### 2.5. Cancer Disease Rates

Analysis of rates for cancer by site was conducted using the Utah Cancer Registry, which was established in 1966, and became an NCI Surveillance, Epidemiology, and End-Results (SEER) registry in 1973. Cohort-specific rates for 15 cancer sites with at least 100 observed cases were calculated by dividing the observed number of cancers of each site in each cohort by the total number of individuals in that cohort.

### 2.6. Estimation of Relative Risk

The RRs for each comorbid condition and each cancer site for all 85,787 individuals identified to be at high risk of AD were estimated using the UPDB population of individuals with linked genealogy and linked hospital data as the observed number of individuals at high-risk for AD with the specific comorbid condition or cancer divided by the expected number of individuals at high-risk for AD with the specific comorbid condition or cancer as described above. Two-sided *p*-values and 95% confidence intervals were derived from the Agresti method [27] and corrected for multiple testing by dividing the calculated significance by the number of statistical tests performed.

For the set of 85,787 individuals identified to be at high-risk for AD, we estimated the RR for 15 common cancers and 5 comorbid conditions previously associated with AD. In addition, we considered association with sex and body mass index (BMI). We compared the sex distribution of the individuals at high-risk for AD to the sex distribution of the 1.8 million individuals in UPDB using Pearson’s chi-squared test. Using the most recent recorded data, BMI for the individuals at high-risk for AD with linked Drivers’ License data (*n* = 44,786) was compared to the UPDB individuals also with linked Drivers’ License data (*n* = 879,177). We used recognized BMI categories (underweight: BMI< 18.5; normal: 18.5 ≤ BMI < 25; overweight: 25.0 ≤ BMI < 30; Obese: 30.0 ≤ BMI < 40; and morbidly obese: BMI ≥ 40) and tested using Pearson’s chi-squared test.

## 3. Results

We first sought to establish and compare the rate of various co-morbidities in high-risk and normal risk cohorts. We found that among people at high-risk for AD, rates of diabetes, hypertension and heart disease were modestly, but significantly lower than the cohort with normal AD risk (Table 1). The estimated relative risk (RR) for these comorbidities were 0.95, 0.97, and 0.95, respectively. We found no difference in the rates of cerebrovascular disease or other dementias between these two groups.

We next turned our attention to the rates of various cancers in the high-risk and normal risk cohorts. In contrast to the previous comorbidities, we found that the rates of all the cancers we examined were higher in the high-risk group relative to the normal risk group (Table 2). After correcting for multiple testing, the increased cancer rate was significant for breast, colorectal, kidney, lung, non-Hodgkin lymphoma, pancreas, stomach, and bladder cancer with estimated RR of 1.23, 1.30, 1.49, 1.25, 1.29, 1.34, 1.59, and 1.40, respectively. The remaining cancer types were also at an increased rate but were not statistically significant after multiple test correction.

We also wanted to explore other differences between the high-risk and normal-risk groups. We found that individuals at high-risk for AD were significantly more likely to be male with 52.1% of the high-risk group relative to 51.4% of the normal risk group being male (Table 3). The high-risk group also had a greater proportion of individuals in the underweight/normal weight category (48.5% of high-risk; 7.2% of normal risk) and a lesser proportion in the overweight/obese/morbidly obese category (51.5% of high-risk; 52.8% of normal risk) (Table 4). This finding was highly significant (2.6 × 10^−8^).

## 4. Discussion

While other studies have analyzed the overlap between AD and these comorbidities, our novel approach has highlighted a different relationship among these diseases. Since the largest risk factor for developing AD is increased age, the lower incidence of hypertension, diabetes, and heart disease makes sense as these comorbidities would generally decrease life span. The healthier the individual, the longer they are likely to live and the larger their risk for AD. There may be more meaningful relationships at play; further research into the pathophysiologic interplay between these diseases and AD is needed.

Most cases of dementia are not clearly defined as a single type, and many risk factors for the various types overlap. The finding that there was no significant relationship between AD and CVD, or AD and non-AD dementia would be expected given the overlap in dementia diagnoses. Additionally, the shared etiologies between the various kinds of dementia make it difficult to identify risk factors or comorbidities specific to each form.

Establishing the exact relationship between various cancers and AD has proven difficult because both diseases have an increased morbidity, which complicates analysis after diagnosis. Specific types of cancer may also differ in their relationship with other diseases relative to other cancers. Previous findings have suggested an inverse relationship between several types of cancer and AD [3,4,5,6,19,20,21]. In contrast to this, we found that a high familial risk of AD had a concomitant increase in risk of all cancers we studied. This increase in risk is significant for many cancer types after correction for multiple testing. This finding might also be explained by otherwise healthy aging. Since age is a risk factor for both AD and many forms of cancer the observed decrease in other comorbidities in our study may explain the increased risk of these specific age-related cancers or be indicative of a more complicated relationship yet to be discovered. The subjects selected for this study have high AD risk but do not have AD, leaving them at a higher risk for age-related cancers without having been diagnosed with AD. According to the National Cancer Institute all statistically significant cancers are strongly associated with increasing risk over time except for prostate; further investigation is needed to establish any sort of causal relationship in this trend.

There are some suggested interactions beyond that of age-related risks. For example, some breast cancer chemotherapeutic treatments were found to increase the risk of subsequent AD diagnosis [16]. Androgen deprivation therapy, a treatment for prostate cancer, has been shown to not increase the risk of subsequent AD diagnosis [17]. Research into PI3K, HSF1, and Pin1 have also suggested that the inverse relationship between AD and cancer stems from a dysregulation in both diseases, but in opposite directions [20,21,22]. Further research into these pathways and their effect on disease development or progression is needed to establish these relationships more firmly.

A recent paper also analyzed the effects of cancer on subsequent AD diagnosis [28]. They found that there was not a significant change in the subsequent development of AD after recovering from cancer when the study is properly controlled. They further explored previous literature looking at this relationship and found that previous studies had not controlled for the increase in morbidity, sample size, or sample sex and age distribution differences. The controls and statistical analysis used in this study are strong and should be used in other settings analyzing groups that have been diagnosed with AD, cancer, and other diseases with increased mortality. Our unique analysis on age matched groups that are at high risk for AD also aims to avoid the potential confounding variable of increased mortality among these populations of interest. We believe our findings do not conflict with the findings of Hanson et al., and provide a unique and important set of data that should shape future research interests on AD, cancer, and the selected morbidities.

The extensive records available through the UPDB strengthen these analyses but the cultural and genetic homogeneity of Utahns also limit them. The similarity in most lifestyle and genetic factors in this population limit the generalizability of these disease rates. Further studies will be needed in more populations to see if the same rates hold true. At the time of the study all the AD diagnoses were removed from the study, however it is possible there were other cases present that had not yet been diagnosed. The size of the population of interest helps to mitigate any complications from this possibility but this does pose another limiting factor. Furthermore, although the size of this population does reduce any confounding factors, there is the possibility that other diseases not studied here may complicate or change the disease rates we found.

## 5. Conclusions

Healthy aging, without complicated medical issues, leaves individuals at higher risk for senescence related diseases later in life. The lack of genetic information on the individuals included in this study limits our ability to generalize the findings, but our findings do support the hypothesis that individuals who have fewer of the comorbid conditions of interest are at higher risk for AD due to age. Our findings show that individuals at high risk for AD have a decreased incidence of various other diseases. Our findings further support the fact that our high-risk group was found to have an increased incidence of various age-related cancers, but no significant correlation with cancers that do not have a large age-risk component. There is the possibility that a more meaningful or etiological relationship exists among these various comorbidities. Further research into the etiological relationship between AD and these comorbidities may elucidate these possible interactions.

It is important to note that our unique approach aimed to remove confounding factors of an active AD diagnosis to draw clearer conclusions; however, real world comorbid diagnoses of AD and hypertension, diabetes, and heart disease may affect the disease interactions and ability to diagnose. Our study aimed to highlight the relationship between AD risk and comorbid occurrence rather than multiple diagnoses. We suggest this as an area of future research along with the etiology of the diseases and their potential overlap.

## Figures and Tables

**Table 1 ijerph-19-16419-t001:** Estimated relative risks for comorbid conditions previously associated with AD, including diabetes, hypertension, heart disease, cerebrovascular disease, stroke, and other dementia for 85,787 individuals identified to be at high-risk of AD.

	Relative Risk	95% Confidence Interval	2-Tailed *p*-Value
Diabetes	0.95	(0.92, 0.97)	1.66 × 10^−4^
Hypertension	0.97	(0.95, 0.99)	5.75 × 10^−4^
Heart Disease	0.95	(0.93, 0.98)	3.99 × 10^−5^
Cerebrovascular Disease	1.02	(0.98, 1.05)	2.96 × 10^−1^
Dementia excluding Alzheimer’s Disease	0.96	(0.92, 1.01)	1.02 × 10^−1^

**Table 2 ijerph-19-16419-t002:** Estimated RRs for 15 common cancers for the 85,787 individuals identified to be at high risk of AD based on family history.

	Relative Risk	95% Confidence Interval	2-Tailed *p*-Value
Any cancer	1.28	(1.24, 1.31)	<1 × 10^−12^
Breast	1.23	(1.16, 1.30)	9.0 × 10^−12^
Colorectal	1.30	(1.21, 1.39)	1.0 × 10^−12^
Kidney	1.49	(1.29, 1.72)	1.8 × 10^−7^
Lung and bronchus	1.25	(1.13, 1.37)	6.9 × 10^−6^
Non-Hodgkin’s lymphoma	1.29	(1.15, 1.44)	1.2 × 10^−5^
Pancreas	1.34	(1.16, 1.55)	8.9 × 10^−5^
Stomach	1.59	(1.36, 1.86)	2.1 × 10^−8^
Bladder	1.40	(1.25, 1.56)	5.9 × 10^−9^
Brain	1.18	(0.97, 1.43)	8.9 × 10^−2^
Cervix	1.05	(0.90, 1.23)	5.2 × 10^−1^
Leukemia	1.26	(1.10, 1.43)	6.5 × 10^−4^
Melanoma	1.16	(1.07, 1.26)	4.0 × 10^−4^
Ovary	1.21	(1.01, 1.43)	3.0 × 10^−2^
Prostate	1.40	(1.33, 1.47)	<1 × 10^−12^
Thyroid	1.18	(1.01, 1.37)	3.4 × 10^−2^

**Table 3 ijerph-19-16419-t003:** Sex distribution for 85,787 individuals at high-risk for AD compared to the over 1.8 million individuals with extensive genealogy linked to Utah founders in the UPDB population (Pearson’s chi-squared test *p* = 1.8 × 10^−5^).

	At High-Risk for AD (% of Total)	UPDB Population (% of Total)
Male	44,733 (52.1%)	937,255 (51.4%)
Female	41,054 (47.9%)	886,364 (48.6%)

**Table 4 ijerph-19-16419-t004:** Body mass index distributions for 44,786 high-risk non-AD subjects at high-risk for AD and 879,177 individuals in UPDB with linked Drivers’ License data and extensive genealogy data (*p* = 2.625 × 10^−8^).

	High-Risk for AD (% of Total)	UPDB Population (% of Total)
Underweight + Normal	21,741 (48.5%)	414,947 (47.2%)
Overweight + Obese + Morbidly Obese	23,045 (51.5%)	464,230 (52.8%)

## Data Availability

Data can only be made available by request to the Utah Population Database.

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
