# Peer review of "Comorbidity and Cancer Disease Rates among Those at High-Risk for Alzheimer’s Disease: A Population Database Analysis"

_ijerph, 2022, doi:10.3390/ijerph192416419_

Round 1
Reviewer 1 Report
This study investigates comorbidity and cancer disease rates in those individuals who were at risk for developing Alzheimer’s Disease based on family history. While, the author needs further to clarify below issues:
1. In the introduction part, the authors mentioned previous literature which demonstrates mixed results between AD and cancers association, with some findings suggesting AD to be less likely to have certain cancers, but others suggesting no relationship. For this research gap, I am doubtful that it can be addressed using a population who have family history of AD but without defined AD diagnosis as presented in this paper.
2. In the methods part, “All individuals belonging to any family history constellation with estimated Relative Risk for AD > 2.0 were considered at high risk for Alzheimer’s disease.” How the cut off 2.0 was selected, any theoretical bases? And which software was used by authors for data analysis? How was the multiple testing done? More detailed information needs to be supplemented.
3. In results, the information of demographics of the study population was missing, for example age, gender, education levels, etc.
4. In discussion, same as in the introduction. The references used are related with cancer and diagnosed AD patients which didn’t compatible with the current population in the study. This study based on a population with high risk for developing Alzheimer’s Disease but without AD diagnosis is a concern for me. I think the study design has significant drawbacks.
Author Response
Reviewer 1
This study investigates comorbidity and cancer disease rates in those individuals who were at risk for developing Alzheimer’s Disease based on family history. While, the author needs further to clarify below issues:
1. In the introduction part, the authors mentioned previous literature which demonstrates mixed results between AD and cancers association, with some findings suggesting AD to be less likely to have certain cancers, but others suggesting no relationship. For this research gap, I am doubtful that it can be addressed using a population who have family history of AD but without defined AD diagnosis as presented in this paper.
The conflicting findings of disease comorbidity in the AD literature is precisely what motivated our study design. Trying to investigate comorbid conditions in individuals who have the confounding diagnosis of AD is likely to have led to so many conflicting findings. Given our unique resources, we can investigate this issue in a different way; we have rather attempted to investigate the association of comorbid conditions in individuals who are highest risk of AD, thus limiting the noise from a concordant AD diagnosis. We discuss this study design and the motivation and advantages of the design in some detail in the INTRODUCTION.
2. In the methods part, “All individuals belonging to any family history constellation with estimated Relative Risk for AD > 2.0 were considered at high risk for Alzheimer’s disease.” How the cut off 2.0 was selected, any theoretical bases? And which software was used by authors for data analysis? How was the multiple testing done? More detailed information needs to be supplemented.
A RR > 2.0 is very commonly used in epidemiologic studies to designate individuals at highest risk of a disorder, and we have used this common cut off here.
We describe the exact methods used for data analysis in significant detail in METHODS; we developed these simple statistical measures and did not use any outside software. We have tried to add more clarification in METHODS. We have added details on the precise method for multiple testing correction.
3. In results, the information of demographics of the study population was missing, for example age, gender, education levels, etc.
We have added some sentences generally describing the Utah population represented in the UPDB.
4. In discussion, same as in the introduction. The references used are related with cancer and diagnosed AD patients which didn’t compatible with the current population in the study. This study based on a population with high risk for developing Alzheimer’s Disease but without AD diagnosis is a concern for me. I think the study design has significant drawbacks.
See response to #1 above. We respectfully disagree and feel that our study uses an innovative approach to deal with a difficult issue.
Reviewer 2 Report
Thank you for the opportunity to review the paper entitled “Comorbidity and cancer disease rates among those at high-risk for Alzheimer's disease: A population database analysis.” This study used data from the Utah Population Database to examine people at high-risk for AD based on family history of AD but who did not yet have a diagnosis of AD (n=75,877). Interestingly, people who were at risk for AD had a lower incidence of diabetes, hypertension, and heart disease but not differences in other types of cerebrovascular disease or other forms of dementia. Conversely, there was an excess incident of breast cancer, colorectal cancer, kidney cancer, lung, non-Hodgkin’s Lymphoma cancer, pancreas cancer, stomach cancer, and bladder cancer.
On line 58, I think there should be a comma not a period before the “but”.
Introduction
The introduction does an excellent job summarizing previous studies and their limitations.
Methods
Please list all comorbidities considered in Line 132. Please consider adding the relevant ICD-9/10 codes as a supplemental document.
Results
Can a table be provided to better describe the demographics and prevalence of diseases between the at-risk group and the normal risk group?
Discussion
Please provide a citation for the sentence listed in line 221: “Previous findings have suggested an inverse relationship between several types of cancer and AD.”
Although beyond the scope of the current project, I hope the authors will consider discussing the prevalence of comorbidities in people with diagnoses of AD. I am curious whether having comorbidities lowers one’s likelihood of receiving an actual AD diagnosis or not. Many diagnose AD through a sequential process of exclusion, especially excluding people who may have “other causes” of cognitive decline (Fleming et al., 1995). What is the authors’ take on this process based on current literature and their current findings? How do biomarkers change how we think about the process of exclusion? Surely if many health conditions increase AD risk, how would they be expected to not co-occur at time of diagnosis? Would confirmatory biomarkers allow for us to move beyond diagnoses based on exclusion due to comorbidities?
Reference:
Fleming, K. C., Adams, A. C., & Petersen, R. C. (1995, November). Dementia: diagnosis and evaluation. In Mayo Clinic Proceedings (Vol. 70, No. 11, pp. 1093-1107). Elsevier.
Author Response
Reviewer 2
Thank you for the opportunity to review the paper entitled “Comorbidity and cancer disease rates among those at high-risk for Alzheimer's disease: A population database analysis.” This study used data from the Utah Population Database to examine people at high-risk for AD based on family history of AD but who did not yet have a diagnosis of AD (n=75,877). Interestingly, people who were at risk for AD had a lower incidence of diabetes, hypertension, and heart disease but not differences in other types of cerebrovascular disease or other forms of dementia. Conversely, there was an excess incident of breast cancer, colorectal cancer, kidney cancer, lung, non-Hodgkin’s Lymphoma cancer, pancreas cancer, stomach cancer, and bladder cancer.
-On line 58, I think there should be a comma not a period before the “but”.
Thank you; fixed.
-Introduction-The introduction does an excellent job summarizing previous studies and their limitations.
Thank you.
-Methods-Please list all comorbidities considered in Line 132. Please consider adding the relevant ICD-9/10 codes as a supplemental document.
Good idea; we have added the list of comorbid conditions considered as well as the relevant ICD codes.
-Results- Can a table be provided to better describe the demographics and prevalence of diseases between the at-risk group and the normal risk group?
As described, we did not have a “normal risk group” for comparison. Rather, we estimated cohort-specific disease rates for each condition for the appropriate UPDB population, using all individuals with linked genealogy, and linked hospital diagnosis data as described. Cohorts were defined by sex and 5-year birth year range and birth state, as described. We have added clarification in METHODS descriptions.
Discussion
Please provide a citation for the sentence listed in line 221: “Previous findings have suggested an inverse relationship between several types of cancer and AD.”
We have added the references discussing the AD and cancer association that were previously discussed in the INTRODUCTION.
Although beyond the scope of the current project, I hope the authors will consider discussing the prevalence of comorbidities in people with diagnoses of AD. I am curious whether having comorbidities lowers one’s likelihood of receiving an actual AD diagnosis or not. Many diagnose AD through a sequential process of exclusion, especially excluding people who may have “other causes” of cognitive decline (Fleming et al., 1995). What is the authors’ take on this process based on current literature and their current findings? How do biomarkers change how we think about the process of exclusion? Surely if many health conditions increase AD risk, how would they be expected to not co-occur at time of diagnosis? Would confirmatory biomarkers allow for us to move beyond diagnoses based on exclusion due to comorbidities?
The reviewer raises very interesting issues that we agree are generally beyond the scope of our project. Our expertise is with using the unique Utah data resources to identify those at highest risk of defined phenotypes, based on their family history, and more typically, to identify the shared genetic factors (variants) among the related, affected individuals who are members of high-risk pedigrees. We have had some small success with both AD, and with healthy longevity using Utah resources.
Reference:
Fleming, K. C., Adams, A. C., & Petersen, R. C. (1995, November). Dementia: diagnosis and evaluation. In Mayo Clinic Proceedings (Vol. 70, No. 11, pp. 1093-1107). Elsevier.
Round 2
Reviewer 1 Report
The authors have clarified my concerns and made some improvements.